# The Effect of an α-7 Nicotinic Allosteric Modulator PNU120596 and NMDA Receptor Antagonist Memantine on Depressive-like Behavior Induced by LPS in Mice: The Involvement of Brain Microglia [note 1]

**DOI:** 10.3390/brainsci12111493

**Published:** 2022-11-03

**Authors:** Sami Alzarea, Muzaffar Abbas, Patrick J. Ronan, Kabirullah Lutfy, Shafiqur Rahman

**Affiliations:** 1Department of Pharmaceutical Sciences, College of Pharmacy, South Dakota State University, Brookings, SD 57007, USA; 2Research Service, Sioux Falls VA Healthcare System, Sioux Falls, SD 57105, USA; 3Department of Psychiatry and Basic Biomedical Sciences, University of South Dakota Sanford School of Medicine, Sioux Falls, SD 57105, USA; 4College of Pharmacy, Western University of Health Sciences, Pomona, CA 91766, USA

**Keywords:** nicotinic receptor, major depressive disorder, neuroinflammation, α7 nicotinic receptor positive allosteric modulator, N-methyl-D-aspartate receptor, mice

## Abstract

Nicotinic acetylcholine receptors (nAChRs), particularly the α7 nAChR, play a critical role in neuroinflammation and microglial activation associated with major depressive disorder (MDD). Microglial quinolinic acid (QUIN), which is synthesized by 3-hydroxyanthranilic acid dioxygenase (HAAO), is an N-methyl-D-aspartate (NMDA) receptor agonist and has been implicated in the development of MDD-related symptoms. In the present study, we assessed the effects of PNU120596, an α7 nAChR positive allosteric modulator (PAM), on HAAO expression and QUIN formation in the hippocampus and prefrontal cortex. We also investigated the effects of memantine, an NMDA receptor antagonist, alone and in combination with PNU120596 on cognitive deficit and depressive-like behaviors induced by lipopolysaccharide (LPS) in mice using the Y-maze and forced swim test, respectively. LPS (1 mg/kg, i.p.) elevated HAAO expression and QUIN formation in the hippocampus and prefrontal cortex, which were reduced with pretreatment with PNU120596 (4 mg/kg, i.p.). Furthermore, memantine (1 or 3 mg/kg, i.p.) prevented the cognitive deficit and depressive-like behaviors induced by LPS in mice. Together, these results suggest that the antidepressant-like effects of PNU120596 are mediated by attenuation of LPS-induced QUIN formation. Therefore, α7 nAChR PAM could be a potential therapeutic candidate for MDD associated with neurotoxic glutamatergic transmission.

## 1. Introduction

Major depressive disorder (MDD) is one of the leading causes of disability and affects more than 4% of the population worldwide [1,2,3]. A growing body of literature suggests that MDD is a multifactorial disease with numerous causes, including genetic susceptibility, stress and/or other pathological processes involving several key brain areas, such as the prefrontal cortex, hippocampus, amygdala, and nucleus accumbens [1,2]. Although the precise cause of MDD remains unidentified, peripheral inflammation might increase the risk of MDD in a subgroup of the patient population [3,4]. Peripheral inflammation has been suggested to promote microglia activation, resident immune cells in the brain, via several pathways [5,6,7]. Cytokines and chemokines, oxidative stress, and local metabolic changes are produced following microglia activation [8]. Furthermore, during neuroinflammation, microglia play a significant role in neurotoxic metabolite quinolinic acid (QUIN) synthesis by activating 3-hydroxyanthranilic acid dioxygenase (HAAO) [9,10,11]. QUIN, known as an an N-methyl-D-aspartate (NMDA) receptor agonist has been implicated in MDD induced by neuroinflammation [10,11]. In addition, patients with interferon-α had elevated brain QUIN levels and showed MDD-related symptoms, showing a relationship between MDD and brain QUIN [12]. In addition, patients suffering from depression and having suicide attempts also had elevated brain QUIN levels [13], indicating microglial QUIN as a potentially important target for modulating MDD symptoms.

Emerging evidence has indicated the involvement of α7 nAChRs in neuroinflammation and microglial activation in the brain [14,15,16]. The activation of α7 nAChRs on microglia promotes mobilization of intracellular calcium in microglia [17,18,19]. However, the anti-inflammatory signaling pathways activated by this receptor are independent of transient changes in intracellular calcium, suggesting the dual ionotropic/metabotropic nature of microglial α7 nAChRs [19,20,21]. However, the pharmacological modulation of α7 nAChR on QUIN production remains to be examined. 

The α7 nAChR is a homomeric structure with two types of binding sites to regulate the receptor function: orthosteric and allosteric [20,22,23]. Several selective α7 nAChR agonists acting on the orthosteric site have been reported to reduce inflammation and depression-like behaviors in laboratory mice [24,25]. When activated by orthosteric ligands, these receptors undergo desensitization and limit the receptor function [22]. Evidence suggests that α7 nAChR PAMs could be an appropriate strategy to prevent desensitization and enhance cholinergic transmission [22,26,27]. 

We have previously demonstrated that PNU120596, an α7 nAChR PAM, reduced depression-like behaviors in LPS-induced models of MDD. The antidepressant-like effects of PAM were found to be linked to microglial activation in the hippocampus and prefrontal cortex [15,18], brain areas involved in depressive symptoms [28]. We hypothesize that PNU120596 by inhibiting microglial activation regulates QUIN formation and reduces depression-like behaviors in mice. Therefore, we examined the effects of the PAM on HAAO expression and QUIN formation in the brain regions implicated in depressive symptoms [28] in a mouse model of LPS-induced MDD. We also determined the effects of neurotoxic transmission blockade by using memantine, an NMDA receptor antagonist, on depressive-like behavior induced by LPS, an inflammatory model of MDD. The effect of PNU120596 in conjunction with memantine was also evaluated in this study. 

## 2. Materials and Methods

### 2.1. Animals 

Male C57BL/6J mice (10–12 weeks of age) were purchased from Jackson Laboratory (Bar Harbor, ME, USA). Mice were housed (5 per cage) with free access to food and water in groups of five in standard shoebox cages (29 × 18 × 12 cm) under standard laboratory conditions (22 ± 2 °C, relative humidity 50–60%). Mice were maintained on a 12 h light/dark cycle with lights on at 6 AM. The behavioral experiments were conducted between 9 AM–4 PM. Mice were brought to the test room and allowed to habituate to the room with dim lighting for at least 1 h. A counterbalanced design was used to control any order effects. All experimental procedures described here follow were according to the National Institutes of Health Guide for the Care and Use of Laboratory Animals in Research and were approved by the Institutional Animal Care and Use Committee at South Dakota State University. Good Laboratory Practice and ARRIVE guidelines were obeyed. All efforts were made to ensure minimal animal suffering.

### 2.2. Drugs and Treatment

LPS of Escherichia coli (serotype 0127: B8) and methyllycaconitine (MLA), an α7 nAChR antagonist, were purchased from Sigma-Aldrich (St. Louis, MO, USA). PNU120596 and memantine were purchased from Tocris Bioscience (Ellisville, MO, USA). MLA and memantine were dissolved in normal saline (0.9% NaCl), whereas PNU120596 was reconstituted in saline containing 5% dimethyl sulfoxide and 5% Solutol (Sigma, St. Louis, MO, USA). All drugs were injected intraperitoneally in a volume of 10 ml/kg of body weight. 

### 2.3. Experimental Procedure 

Mice were injected with PNU120596 (1 or 4 mg/kg, i.p.) and 30 min later with LPS (1 mg/kg), as described previously [15,18]. Where we studied the effect of the combination of the twol drugs, MLA (3 mg/kg) was injected 10 min prior to PNU120596 (4 mg/kg) followed, 30 min later, by LPS. Mice were euthanized 24 h later, and the brain of each mouse collected for biochemical analyses. We selected this time point because depressive-like behavior is displayed at this time and has been shown to be prevented by PNU120596 [15,18,29]. The hippocampi and prefrontal cortices were dissected using the Allen Brain Atlas and mouse brain stereotaxic coordinates and stored at −80 °C until further analysis. For behavioral studies, mice were simultaneously treated with memantine (1 or 3 mg/kg) and LPS to determine the antidepressant-like effects of memantine 24 h after LPS administration. The effect of the combination of PNU120596 and memantine on depressive-like behavior induced by LPS was determined using the tail suspension test (TST) and forced swim test (FST). Y-maze was also used to measure any changes in cognition.

### 2.4. Quantitative Real-Time Polymerase Chain Reaction (rt-PCR) 

Quantitative rt-*PCR* was according to earlier reports [14,17]. Total RNA was extracted from 1-mm coronal sections of hippocampal or prefrontal cortical tissue using trizole reagent (Invitrogen, Carlsbad, CA, USA) based on the protocol described by the manufacturer. The cDNA was synthesized using High-Capacity cDNA Reverse Transcription Kit (Applied Biosystems, Carlsbad, CA, USA) and Master Cycler Personal (Eppendorf, Hauppauge, NY, USA), as described previously [14,17]. Primers were purchased from Integrated DNA Technologies (Coralville, IA, USA). The sequence of each primer is provided in Table 1. The cycle threshold (Ct) value was recorded for each gene to calculate their relative expression using delta-delta Ct method.

### 2.5. Immunofluorescence Assay

The protocol for immunofluorescence assays was according to an earlier report [15]. Briefly, 14 µM coronal slices were washed with phosphate-buffered saline (PBS) and incubated in 0.01 M citrate buffer (pH 6.0) at 90 °C in a water bath for 10 min. Hippocampal and corticla sections were then blocked and incubated with a primary antibody against HAAO (1:50, Bioss Inc., Woburn, MA, USA), or conjugated QUIN (1:150, Abcam, Cambridge, MA, USA) overnight (4 °C). Tissue sections were then incubated with a secondary antibody labeled with fluorescein isothiocyanate (FITC) (Santa Cruz Biotechnology, Dallas, TX, USA). The slides were mounted with a mounting medium containing 4′,6′-diamidino-2-phenylindole (DAPI) for nuclear staining and prolonged anti-fade reagent (Santa Cruz Biotechnology, Dallas, TX, USA). A laser scanning confocal microscope (Olympus Fluoview FV1200) was used to measure fluorescence levels. The density of protein immunoreactivity was carried out using Image J software (NIH, Bethesda, MD, USA) as described previously [15].

### 2.6. Behavioral Tests

Locomotor activity: Total distance traveled, used as a measure of locomotor activity, was assessed as described previously [30]. Briefly, mice were individually placed in the periphery of the cage (40 cm × 40 cm × 35 cm) and allowed to explore the chamber for 15 min. The first five minutes were used to habituate animals to the chambers, whereas the last 10 min were the test. A camera was vertically positioned 100 cm above the activity chambers to record the animal’s behavior. The videos were then analyzed by a blind experimenter to measure the total distance traveled (m) by each mouse using the ANY-maze software (Stoelting Co., Wood Dale, IL, USA).

Y-maze: The Y-maze was according to a previous study [10]. Briefly, the Y-maze apparatus was made of gray Plexiglas and consisted of three closed arms (35 cm × 5 cm × 10 cm) at a 120° angle from each other. Each mouse was placed at the center of the maze and allowed to freely explore the three arms for eight min. The arm entry of each mouse was videorecorded, and the videos were analyzed to measure spontaneous alternations. An arm entry was defined as the presence of all four paws in one arm. The alternations were counted when a mouse visited three different arms without making a return entry to an already visited arm. The percent spontaneous alternations were calculated using the formula (alternate arm entries/total number of entries) × 100. At the end of session, the apparatus was thoroughly cleaned with 70% ethanol after the removal of each mouse.

Tail suspension test: The protocol was according to previous reports [30,31]. Briefly, each mouse was suspended by the tail 45-cm from the floor by attaching the tail to a hook of the testing chamber at one cm from the tip of the tail using a medical adhesive tape. The test was conducted for six min. The videos were analyzed to measure immobility time for each mouse, defined by the absence of paw or body movements as immobility.

Forced swim test: The prcedure was according to previous studies [31,32]. Briefly, mice were placed individually in a cylindrical (Wood Dale, IL, USA) acrylic tank (45 cm high × 20 cm diameter), which was filled (25 cm) with water (25 ± 1 °C), and allowed to swim for 6 min. The activity of each mouse was videorecorded, and the videos were analyzed to measure immobility time. Immobility was defined as the lack of any activity except those required to keep the head above water. Mice were removed from the cylinder immediately after the test session, gently dried with paper towels, and returned to their home cages.

### 2.7. Statistical Analyses

Data represent mean ± SEM of the amount of time that mice remained immmobile or changes in gene expression. Gene expression and QUIN formation were analyzed using a one-way analysis of variance (ANOVA). Two-way ANOVAs were conducted (LPS vs. control × treatments) for each behavioral test. The post hoc Tukey’s tests were used to compare different groups. All data analyses were performed using GraphPad Prism (GraphPad Inc., San Diego, CA, USA). A difference of *p* < 0.05 between the two groups was considered significant. 

## 3. Results

### 3.1. Effects of PNU120596 on HAAO Expression in the Hippocampus and Prefrontal Cortex

We measured the expression of HAAO, the rate-limiting enzyme for the production of QUIN, in hippocampal and cortical slices to determine if this could account for the protective effect of PNU120596 on depressive-like behavior induced by LPS [15]. Data analyses indicated that PNU120596 reduced HAAO expression (Figure 1A) in the hippocampus (F_4,23_ = 6.420; *p* < 0.01) and prefrontal cortex (F_4,23_ = 5.583; *p* < 0.01). The post hoc analysis revealed that LPS (1 mg/kg) significantly increased HAAO expression in both brain regions compared to the control group. On the other hand, treatment with PNU120596 (4 mg/kg) significantly decreased HAAO expression in the hippocampus and prefrontal cortex compared to the LPS-treated group. In contrast, MLA significantly blocked the inhbitory effect of PNU120596 on the HAAO expression in both brain regions.

To further evaluate the effects of PNU120596 on the LPS-induced QUIN production, HAAO immunoreactivity was quantified in the dentate gyrus (DG), and cornu ammonis region (CA1 regions of the hippocampus), and prefrontal cortex (Figure 1B). Data analyses revealed that PNU120596 significantly decreased HAAO expression in DG (F_2,12_ = 8.569; *p* < 0.01), CA1 (F_2,12_ = 7.683; *p* < 0.01) and prefrontal cortex (F_2,12_ = 9.172; *p* < 0.01). The *post hoc* test revealed that LPS (1 mg/kg) significantly increased HAAO expression than the control group. Furthermore, PNU120596 (4 mg/kg) significantly reduced the LPS-elevated HAAO expression in all these regions (*p* < 0.05).

### 3.2. Effects of PNU120596 on QUIN Immunoreactivity in DG and CA1 Regions of the Hippocampus and Prefrontal Cortex

To determine the effects of PNU120596 on LPS-induced neurotoxic signal, QUIN immunoreactivity (QUIN-IR) was examined in the DG, CA1, and prefrontal cortex (Figure 2). One-way ANOVA revealed that PNU120596 reduced QUIN-IR in DG (F_2,15_ = 5.080; *p* < 0.05), CA1 (F_2,15_ = 7.539; *p* < 0.01) and prefrontal cortex (F_2,15_ = 5.024; *p* < 0.05). The *post hoc* analysis of the data indicated that LPS (1 mg/kg) significantly increased QUIN-IR than the control group. Moreover, PNU120596 (4 mg/kg) clearly decreased QUIN-IR in all these regions compared to the LPS-treated group.

### 3.3. Effects of Memantine on LPS-Induced Depressive-like Behavior 

Figure 3B depicts the effects of memantine (1 or 3 mg/kg) on the total distance traveled in mice treated with vehicle or LPS. Two-way ANOVA indicated that memantine did not alter locomotor activity in mice of either group (F_2,31_ = 0.07413; *p* = 0.9287). Figure 3C illustrates the effects of memantine on spontaneous alternations for cognitive deficit-like behavior in Y-maze 25.5 h after LPS injection. Two-way ANOVA indicated that memantine significantly (F_2,32_ = 3.471; *p* < 0.05) reversed LPS-mediated reduction in spontaneous alternations. Furthermore, the effects of memantine on immobility time in the TST and FST 27 and 28.5 h, respectively, following LPS treatment are shown in Figure 3D,E. Two-way ANOVA revealed that memantine significantly reversed the LPS-induced increase in immobility time in TST (F_2,32_ = 3.657; *p* < 0.05) and FST (F_2,32_ = 3.478; *p* < 0.05).

### 3.4. Combinatorial Effects of PNU120596 and Memantine on LPS-Induced Depressive-like Behavior 

Figure 4B illustrattes the effects of PNU120596 (1 mg/kg) in conjunction with memantine (1 mg/kg) on locomotor activity 24 h after LPS treatment. Two-way ANOVA suggested that the total distance traveled was not significantly different between the treatment groups (F_2,31_ = 0.3597; *p* = 0.7008). The effect of PNU120596 and memantine together on spontaneous alternations in Y-maze 25.5 h following LPS treatment are shown in Figure 4C. Two-way ANOVA revealed that PNU120596 and memantine coadministration significantly (F_2,32_ = 4.209; *p* < 0.05) prevented the LPS-induced decrease in spontaneous alternations. The combinatorial effects of PNU120596 and memantine on immobility time in TST and FST 27 and 28.5 h, respectively, after LPS injection are shown in Figure 4D,E. Two-way ANOVA showed that PNU120596 in conjunction with memantine significantly reduced LPS-induced increase in immobility time in TST (F_2,32_ = 3.424; *p* < 0.05) and FST (F_2,36_ = 5.338; *p* < 0.01). PNU 120596 (1 mg/kg) administration alone after LPS injection did not alter behavioral measures in the Y-maze, TST, and FST.

## 4. Discussion

In the present study, LPS treatment significantly increased in HAAO expression and QUIN formation in the hippocampus and prefrontal cortex. These changes were blocked by pretreatment with PNU120596 in a dose-dependent fashion. Memantine reduced LPS-induced depressive-like behavior observed 24 h after LPS administration. Memantine also attenuated LPS-induced cognitive deficit-like behaviors. Additionally, memantine increased the effects of PNU120596 against LPS-induced depressive-like behavior. This was evident when mice were injected with PNU120596 and memantine together. Systemic LPS injection causes inflammation in the CNS associated with changes in sickness behavior [5]. These changes lead to depressive-like behavior 24 h after peripheral LPS injection [5,29]. QUIN produced by HAAO in the brain after LPS administration has been reported to induce depressive-like behavior in mice [10,11]. Our results demonstrated that LPS increased HAAO expression and QUIN formation in the hippocampus and prefrontal cortex likely via microglia activation, as microglia are the primary source of QUIN [9].

Recent studies have reported that genetic and pharmacological approaches targeting HAAO or QUIN regulate depressive-like behavior in mice. For example, mice lacking HAAO did not develop depressive-like behavior after LPS injection. On the other hand, hydroxykynurenine, a precursor of QUIN, treatment induced depressive-like behavior. Alaterations in depressive-like behavior were proposed to be dependent on QUIN formation [10]. Additionally, antagonism of the NMDA receptor by ketamine reduced LPS-induced depressive-like behavior in mice [11]. Thus, we investigated the effects of PNU120596 on LPS-induced HAAO expression and QUIN formation in the hippocampus and prefrontal cortex. Our results revealed that LPS upregulated HAAO expression and increased QUIN formation, effects prevented by PNU120596. The consistent results between HAAO mRNA and protein indicate that the effects of PNU120596 on HAAO depend on the activation of α7 nAChR since MLA, an α7 nAChR antagonist, was able to block these changes. These data suggest a relationship between the inhibitory effects of PNU120596 on microglial activation [15,18] and the formation of QUIN in the hippocampus and prefrontal cortex. Previous studies indicated that microglia, specifically in the hippocampus, respond to inflammatory signals [33]. The early-life disconnection of the ventral hippocampus was shown to regulate microglial function in brain regions connected with hippocampus, including the prefrontal cortex [34]. These may be the reasons for QUIN formation in those brain regions. Therefore, PNU120596 likely reduced the neurotoxic QUIN signal due to its inhibitory actions on microglial activation in the hippocampus and prefrontal cortex. These effects may reduce QUIN formation by preventing the increased activity of HAAO. Furthermore, pharmacological activation of α7 nAChR results in neuroprotective effects against the neurotoxic signal of QUIN [35]. We offer that PNU120596 may act beyond the regulation of microglial activation to provide neuronal protection against LPS-induced neuronal excitotoxicity in the hippocampus and prefrontal cortex. 

QUIN is a part of the brain kynurenine pathway of tryptophan metabolism [29]. During neuroinflammation, proinflammatory cytokines lead to the activation of indoleamine 2,3-dioxygenase that shifts brain serotonin synthesis to kynurenine [29]. The kynurenine is converted to kynurenic acid, an NMDA receptor antagonist, under normal conditions. Conversely, the metabolism of kynurenine is shifted to form QUIN in response to LPS-induced inflammation in the brain [11,36]. Therefore, PNU120596 likely prevents tryptophan breakdown to increase serotonin levels. In this case, we propose that PNU120596 might shift the pathway to synthesize more kynurenic acid to prevent the neurotoxic effects of QUIN. However, further research may be needed to provide support for this notion. The α7 nAChRs expressed on microglia are responsible for QUIN formation [9,16], believed to depend on neuroinflammatory signals [29,37]. LPS-induced depressive-like behavior is likely to be mediated by a neurotoxic signal produced by QUIN-mediated activation of NMDA receptor. To test this possibility, we used memantine, an NMDA receptor antagonist, to assess if LPS-mediated depressive-like behavior is mediated by the stimulation of the glutamatergic neurotransmission. Our results revealed that memantine significantly attenuated LPS-induced depressive-like behavior. Previous studies have demonstrated that memantine reduces depressive-like behavior in other models [38,39]. Memantine was shown to increase the level of neurotransmitters implicated in mood disorders, such as serotonin, norepinephrine, and dopamine in the brain at a basal condition [40]. We did not observe any antidepressant-like effects of memantine at the basal level, suggesting that memantine might not be able to enhance brain neurotransmitters to produce antidepressant-like effects. However, memantine significantly reduced LPS-induced depressive-like behavior linked with increased QUIN production. Therefore, the antidepressant-like effects of memantine are likely to be specific to NMDA receptor blockade in this mouse model of LPS-induced depression-like behaviors.

Prior research has reported that the hippocampus regulates spatial memory [10,41,42], and the prefrontal cortex has a role in reversal learning [43,44]. Consistent with the previous studies, LPS induces cognitive deficit-like behavior, suggesting that QUIN may cause this via mechanism described above [10]. In the present study, we found that memantine decreased LPS-induced cognitive deficits in mice. These results suggest that memantine prevents QUIN’s deleterious actions on the hippocampus and prefrontal cortex. PNU120596 exerted antidepressant-like actions against LPS-induced depressive-like behavior [15]. With respect to the interaction between α7 nAChR and NMDA receptor, we discovered that coadministration of ineffective doses of PNU120596 (1 mg/kg) and memantine (1 mg/kg) blocked the LPS-induced cognitive deficit and depressive-like behaviors. These data suggest that the two drugs may provide a partial anti-inflammatory signal partly due to blockade of α7 nAChR and in part by inhibition of the neurotoxic glutamatergic signal induced by QUIN.

## 5. Conclusions

In conclusion, these results reveal that the selective α7 nAChR positive allosteric modulator PNU120596 prevents the increased HAAO expression and QUIN formation induced by LPS in the hippocampus and prefrontal cortex (Figure 5). The additive/synergistic effect of Not PNU120596 and memantine may stem from an interaction between blockade of α7 nAChR and NMDA receptor by inducing anti-inflammatory effects and antagonizing the neurotoxic glutamatergic signaling. These effects further our understanding of the mechanisms involved in the antidepressant-like effects of PNU120596. Thus, α7 nAChR PAM could represent a potential therapeutic drug candidate for the management of MDD associated with toxic glutamatergic transmission in the brain. Our data also reveal an interaction between α7 nAChR and NMDA receptor in the pathophysiology of depression-like behavior in an inflammatory mouse model of MDD. Our data indicate that the reduced microglial QUIN formation and neurotoxic glutamatergic-mediated signaling are critically involved in the antidepressant-like effects of PNU120596 in this model of MDD. However, other glial mechanisms and other brain regions associated with MDD may also be involved, which require further investigation for future drug development and treatment strategy for MDD. For example, our studies have revealed that α7 nAChR plays a pivotal role in the inflammatory function of microglia regulating depression-like behavior in mice. However, astrocytes [45,46,47] also play a significant role in the modulation of depression-like behavior. Therefore, future research is needed to determine the role of astrocytes in the hippocampus and prefrontal cortex, as we have determined the biochemical alterations only in the hippocampus and prefrontal cortex in the present study. Other brain areas, such as amygdala [48,49] and nucleus accumbens [50,51], also regulate depression-like behavior. Therefore, further studies are needed to assess the role of these brain regions using a similar MDD model and drug candidates.

## Figures and Tables

**Figure 1 brainsci-12-01493-f001:**
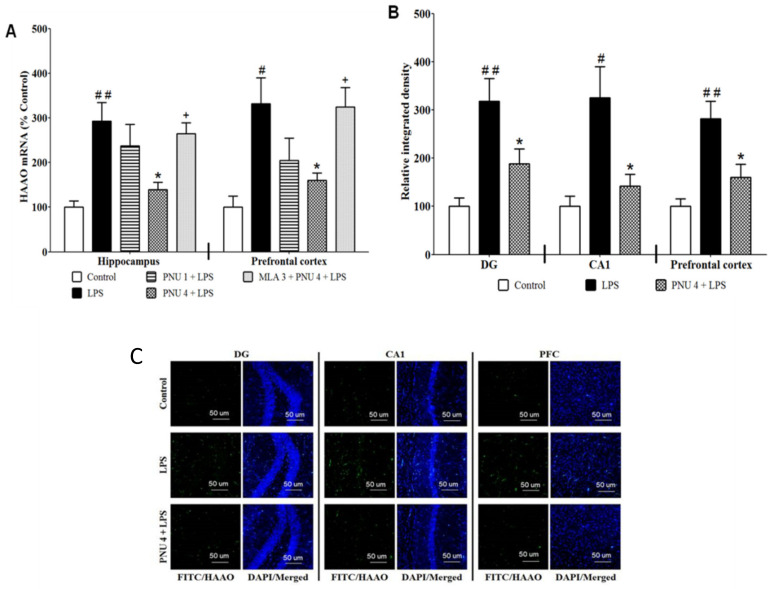
(**A**) PNU120596 (PNU) reduced LPS-elevated HAAO expression in the hippocampus and prefrontal cortex. LPS (1 mg/kg) significantly increased HAAO expression in the hippocampus (*p* < 0.01) or prefrontal cortex (*p* < 0.05) than controls. PNU (4 mg/kg) significantly (*p* < 0.05) reduced HAAO expression in the hippocampus or prefrontal cortex than the LPS-treated group. (**B**) Effects of PNU on the HAAO immunoreactivity in the DG and CA1 regions of the mouse hippocampus and prefrontal cortex. LPS (1 mg/kg) significantly elevated HAAO expression in DG (*p* < 0.01), CA1 (*p* < 0.05) or prefrontal cortex (*p* < 0.01) than control group. PNU (4 mg/kg) significantly (*p* < 0.05) attenuated HAAO expression in the DG, CA1 or prefrontal cortex than the LPS-treated group. (**C**) Representative images of immunofluorescence in DG, CA1, and prefrontal cortex. Magnification 20×, scale bar = 50 μm. ^#^
*p* < 0.05 or ^##^
*p* < 0.01, LPS (1 mg/kg) vs. control; * *p* < 0.05, PNU (4 mg/kg) + LPS vs. LPS alone; **^+^**
*p* < 0.05, MLA (3 mg/kg) + PNU (4 mg/kg) + LPS vs. PNU (4 mg/kg) + LPS. Data are expressed as mean ± SEM (4–7 mice/group).

**Figure 2 brainsci-12-01493-f002:**
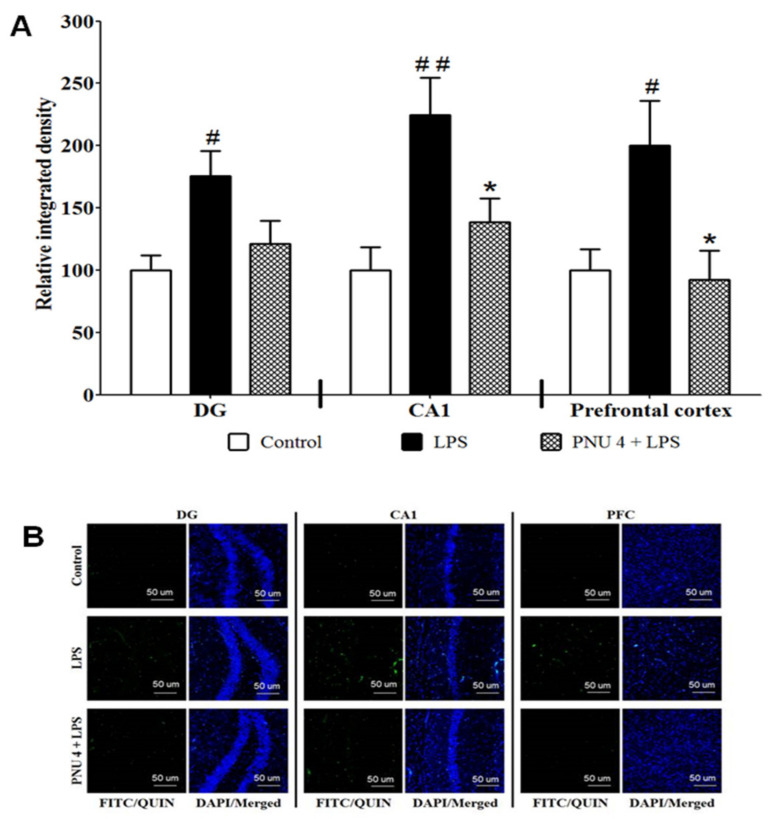
(**A**) Effects of PNU120596 (PNU) on the QUIN-immunoreactivity (QUIN-IR) in DG and CA1 regions of the hippocampus and prefrontal cortex. LPS (1 mg/kg) significantly increased QUIN-IR in DG (*p* < 0.05), CA1 (*p* < 0.01) or prefrontal cortex (*p* < 0.05) compared to control. PNU (4 mg/kg) reduced QUIN in the DG, CA1 or prefrontal cortex than the LPS-treated group. (**B**) Representative immunofluorescence images in DG, CA1, and prefrontal cortex. Magnification 20×, scale bar = 50 μm. ^#^
*p* < 0.05 or ^##^
*p* < 0.01, LPS (1 mg/kg) vs. control; * *p* < 0.05, PNU (4 mg/kg) plus LPS vs. LPS alone. Data are expressed as mean ± SEM (*n* = 6/group).

**Figure 3 brainsci-12-01493-f003:**
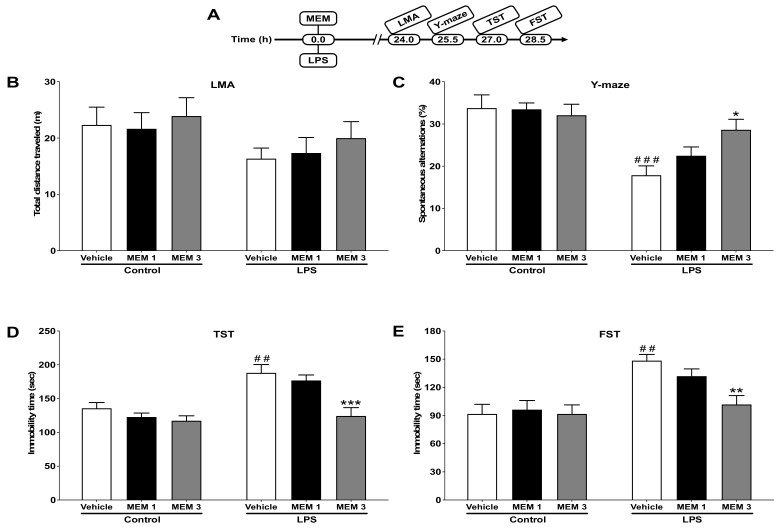
Effects of memantine (MEM, 1 or 3 mg/kg) on LPS-induced depressive-like behavior. (**A**) Experimental timeline for drug administrations and behavioral tests. (**B**) Effects of MEM on locomotor activity. The treatments did not show significant effects on the total distance traveled. (**C**) Effects of MEM on the percent of spontaneous alternations in Y-maze. MEM (3 mg/kg) significantly (*p* < 0.05) enhanced spontaneous alternations than LPS treated mice. (**D**) MEM (3 mg/kg) significantly (*p* < 0.001) prevented LPS-induced increase in immobility time in the TST. (**E**) MEM significantly (*p* < 0.01) prevented the LPS-induced increase in immobility time in the FST. ^##^*p* < 0.01 or ^###^
*p* < 0.001, vehicle/LPS (1 mg/kg) vs. vehicle/control; * *p* < 0.05, ** *p* < 0.01 or *** *p* < 0.001, MEM (3 mg/kg) plus LPS vs. vehicle/LPS. Data are expressed as mean ± SEM (*n* = 6–7/group).

**Figure 4 brainsci-12-01493-f004:**
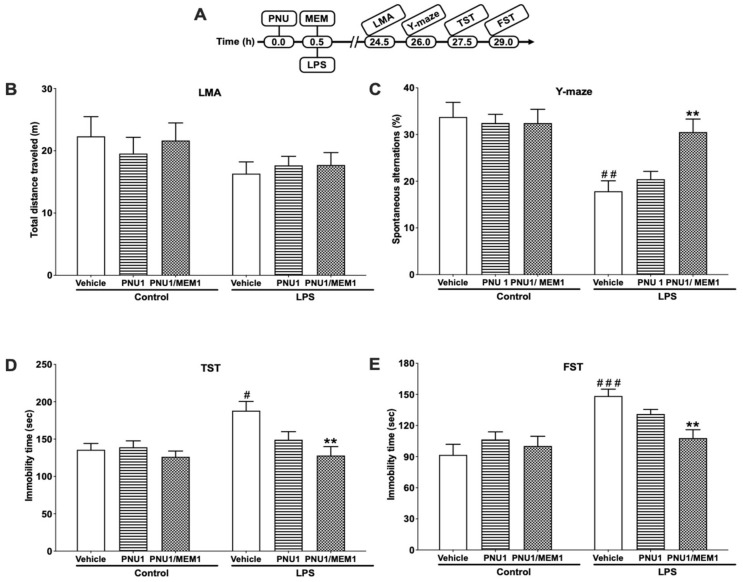
The effects of PNU120596 (PNU, 1 mg/kg) in combination with memantine (MEM, 1 mg/kg) on LPS-induced depressive-like behavior. (**A**) Experimental timeline for drug administrations and behavioral assays. (**B**) The effect of PNU with MEM on total distance traveled (m). The two drugs did not show significantly alter the total distance traveled. (**C**) The effect of PNU with MEM on spontaneous alternations (%) in Y-maze. Treatment with PNU and MEM together significantly (*p* < 0.01) increased spontaneous alternations compared to the LPS-treated group. (**D**) Coadministration of PNU and MEM significantly (*p* < 0.01) decreased immobility time compared to the LPS-treated mice. (**E**) Coadministration of PNU and MEM significantly (*p* < 0.01) decreased immobility time in the FST than the LPS-treated mice. ^#^
*p* < 0.05, ^##^
*p* < 0.01 or ^###^
*p* < 0.001, vehicle/LPS (1 mg/kg) vs. vehicle/control; ** *p* < 0.01, PNU (1 mg/kg) + MEM (1 mg/kg)/LPS vs. vehicle/LPS. Data are expressed as mean ± SEM, *n* = 6–10/group.

**Figure 5 brainsci-12-01493-f005:**
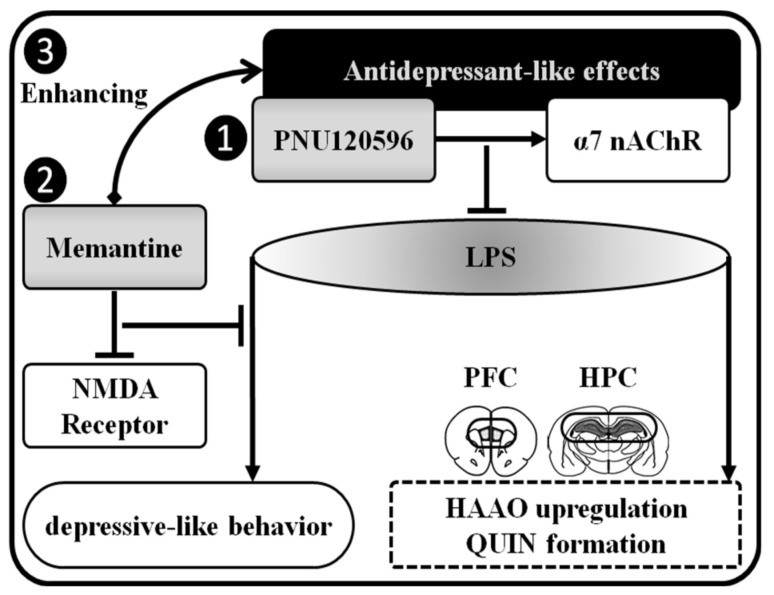
A schematic diagram of how LPS may induce depression-like behaviors and how the α7 nAChR and NMDA receptor antagonists together exert antidepressant-like effects in LPS-mediated inflammatory model of MDD. (1) PNU120596 positively modulates the α7 nAChR via the allosteric binding site. This activation prevents LPS-induced increase in HAAO expression and QUIN formation in the hippocampus (HPC) and prefrontal cortex (PFC). (2) Inhibition of the NMDA receptor-mediated glutamatergic neurotransmission, induced by QUIN, by memantine reduced LPS-induced depression-like behavior. (3) Memantine enhances the antidepressant-like effects of PNU120596, suggesting the interaction between α7 nAChR and NMDA receptor mediating depression-like behavior in mice. →: stimulation; ─┤: inhibition.

**Table 1 brainsci-12-01493-t001:** Sequence of primers used in the current investigation in the quantitative real-time polymerase chain reaction.

Gene	Primer Sequence (5′–3′)
HAAO	GGCTGGTGATTGAGAGAAGG (forward) GGTCCTTACAGTGGAACCATT (reverse)
GAPDH	GTGGAGTCATACTGGAACATGTAG (forward) AATGGTGAAGGTCGGTGTG (reverse)

## Data Availability

All data included in this study are available upon request from the corresponding authors.

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
