# Peer review of "The Effect of an α-7 Nicotinic Allosteric Modulator PNU120596 and NMDA Receptor Antagonist Memantine on Depressive-like Behavior Induced by LPS in Mice: The Involvement of Brain Microglia†"

_brainsci, 2022, doi:10.3390/brainsci12111493_

Round 1

Reviewer 1 Report

Alzarea and colleagues in the present article entitled ‘Alpha-7 Nicotinic Allosteric Modulator PNU120596 and NMDA Receptor Antagonist Memantine Prevents LPS-induced Depressive-like Behavior via Brain Microglial Mechanisms in Mice’, investigated the positive effect of modulator PNU120596 in regulating HAAO expression and QUIN formation in the hippocampus and in the prefrontal cortex during neuroinflammation. Authors also examined the effects of memantine, an NMDA receptor antagonist, and pharmacological interaction between PNU120596 and memantine on LPS-induced cognitive deficit and depressive-like behaviors. Results showed that pretreatment with PNU120596 reduced the LPS-induced increase in HAAO expression and QUIN formation in these brain regions. In addition, memantine (1 or 3 mg/kg) prevented LPS-induced cognitive deficit and depressive-like behaviors reflected by decreasing spontaneous alternation in Y-maze and increasing immobility time in tail suspension test and forced swim test, while a subthreshold dose (1 mg/kg) of memantine enhanced PNU120596's effects against LPS-induced cognitive deficit and depressive-like behaviors.

The main strength of this manuscript is that it addresses an interesting and timely question, providing a captivating interpretation and describing the potential effectiveness of the combination therapy of PNU120596 and memantine as a potential therapeutic in the treatment of neurotoxic glutamatergic transmission. In general, I think the idea of this research article is really interesting and the authors’ fascinating observations on this timely topic may be of interest to the readers of Brain Sciences. However, some comments, as well as some crucial evidence that should be included to support the authors’ argumentation, needed to be addressed to improve the quality of the manuscript, its adequacy, and its readability prior to the publication in the present form. My overall judgment is to publish this research article after the authors have carefully considered my suggestions below, in particular reshaping parts of the Introduction and Methods sections by adding more evidence.

Please consider the following comments:

·       Abstract: According to the Journal’s guidelines, the abstract should be a total of about 200 words maximum. Please correct the actual one. Also, I would suggest to not use abbreviations in this section.

·       In general, I recommend authors to use more evidence to back their claims, especially in the Introduction of the paper, which I believe is currently lacking. Thus, I recommend the authors to attempt to deepen the subject of their manuscript, as the bibliography is too concise: nonetheless, in my opinion, less than 50 articles for a research article are insufficient. I suggest the authors to focus their efforts on researching more relevant literature: I believe that adding more studies and reviews will help them to provide better and more accurate background to this study.

·       Introduction: The ‘Introduction’ section is well-written and nicely presented, with a good balance of descriptive text and information about neurobiological mechanisms underlying major depressive disorder (MDD). Nevertheless, I believe that more information about neuroanatomical pathways underlying basis of MDD will provide a more scientific and more accurate background: specifically, I recommend to focus on how depressive disorder may depend on altered activity within the amygdala and the prefrontal cortex, and to provide information on how frontal lobes dysfunctions (and psychophysiological processes that these regions mediate) (doi: 10.1038/nn3084; doi: 10.1038/nrn2648) are often related to altered reward system and fear-anxiety system, leading to this neuropsychiatric disorder (https://doi.org/10.1111/psyp.14122; https://doi.org/10.3389/fpsyt.2019.00914; https://doi.org/10.3390/biomedicines10081897).

·       Recruitment and study design: Data about animals are not adequately explained. Could the authors specify how did they estimate the exact number of mice? Did they use a power analysis?

·       Quantitative real-time polymerase chain reaction: I suggest rewriting this section more accurately. Please provide more information about experimental details while performing qRT-PCR.

·       In my opinion, I think the ‘Conclusions’ paragraph would benefit from some thoughtful as well as in-depth considerations by the authors, because as it stands, it is very descriptive but not enough theoretical as a discussion should be. Authors should make an effort, trying to explain the theoretical implication as well as the translational application of their research.

·       In according to the previous comment, I would ask the authors to include a proper ‘Limitations and future directions’ section before the end of the manuscript, in which authors can describe in detail and report all the technical issues brought to the surface.

·       Figures and Tables: According to the Journal’s guidelines, please add an explanatory caption for each figure/table within the text. Also, I believe that a visual representation of the experiment’s temporal duration.

Overall, the manuscript contains 1 table, 5 figures and 44 references. This manuscript might carry important value describing the potential effectiveness of the combination therapy of PNU120596 and memantine as a potential therapeutic in the treatment of neurotoxic glutamatergic transmission.

I hope that, after these careful revisions, the manuscript can meet the Journal’s high standards for publication. I am available for a new round of revision of this article.

Best regards,

Reviewer

Reviewer 2 Report

The manuscript is generally well-written and the idea of this study is indeed very interesting. The methodology is very well described and the statistics is relevant.

Nevertheless, I would suggest the authors to improve the discussions section since it is too long and contains unrelevant general information. The discussions section should focus more on assessing the existing data in the literature and compare the findings of the present study with those previously reported in the literature.

The conclusions section should also underline the clinical relevance of the present findings.

Author Response

Please see the attchment.

Round 2

Reviewer 1 Report

The authors did an excellent job clarifying the questions I have raised in my previous rounds of review. Currently, this paper is a well-written, timely piece of research and provides an useful summary of the effectiveness of PNU120596 and memantine combination therapy as a potential therapeutic in the treatment of neurotoxic glutamatergic transmission.

Overall, this is a timely and needed work. It is well researched and nicely written, with a good balance between descriptive and narrative text.

I am always available for other reviews of such interesting and important articles.

Thank You for your work.

Author Response

Thank you very much for your comments and the review process. 

Reviewer 2 Report

Thank you for improving the discussions section. Nevertheless, please remove the references from conclusions section and rephrase the conclusions as an interpretations of your own and not citing the literature.

Author Response

Thank you very much for your comment and time in this review process. We have checked all spellings. Concerning your comment, we would like to add that we have included several references based on previous comment from Reviewer 1. Please note that those references are cited to support some relevant facts only. Nevertheless, our concluding comments and interpretations are independent and own without citations. I hope you will find this satisfactory. Again thank you for your understanding.